# Hemotrophic Mycoplasmas—Vector Transmission in Livestock

**DOI:** 10.3390/microorganisms12071278

**Published:** 2024-06-23

**Authors:** Mareike Arendt, Julia Stadler, Mathias Ritzmann, Julia Ade, Katharina Hoelzle, Ludwig E. Hoelzle

**Affiliations:** 1Department of Livestock Infectiology and Environmental Hygiene, Institute of Animal Science, University of Hohenheim, 70593 Stuttgart, Germany; ma.westenhoefer@gmail.com (M.A.); julia.ade@uni-hohenheim.de (J.A.); katharina.hoelzle@uni-hohenheim.de (K.H.); 2Clinic for Swine, Centre for Clinical Veterinary Medicine, Ludwig-Maximilians-Universität München, 85764 Oberschleissheim, Germany; j.stadler@med.vetmed.uni-muenchen.de (J.S.); m.ritzmann@lmu.de (M.R.)

**Keywords:** hemotrophic mycoplasmas, arthropod vectors, transmission

## Abstract

Hemotrophic mycoplasmas (HMs) are highly host-adapted and specialized pathogens infecting a wide range of mammals including farm animals, i.e., pigs, cattle, sheep, and goats. Although HMs have been known for over 90 years, we still do not know much about the natural transmission routes within herds. Recently, it has been repeatedly discussed in publications that arthropod vectors may play a role in the transmission of HMs from animal to animal. This is mainly since several HM species could be detected in different potential arthropod vectors by PCR. This review summarizes the available literature about the transmission of bovine, porcine, ovine, and caprine HM species by different hematophagous arthropod vectors. Since most studies are only based on the detection of HMs in potential vectors, there are rare data about the actual vector competence of arthropods. Furthermore, there is a need for additional studies to investigate, whether there are biological vectors in which HMs can multiply and be delivered to new hosts.

## 1. Introduction

Hemotrophic mycoplasmas (HMs) are small epicellular, pleomorphic, cell wall-less, and uncultivable bacteria. They parasitize the surface of erythrocytes or even invade them and cause erythrocytic deformity and damage [1,2,3]. HMs are highly adapted to their hosts by virtue of their complex nutritional requirements and the induction of persistent infections [2,4,5]. In 1928, both Schilling and Kikuth identified, for the first time, mouse- and dog-specific HMs (*Eperythrozoon coccoides*, *Haemobartonella canis*) [6,7]; since then, species-specific HMs have been described in the blood of several animals. Until now, at least seven species of hemotrophic mycoplasma have been known in livestock, including *Mycoplasma suis*, *M. parvum*, ‘*Candidatus* (*Ca*.) M. haemosuis’, *M. wenyonii*, ‘*Ca*. M. haemobos’, *M. ovis*, and ‘*Ca*. M. haemovis’ [8,9,10,11,12,13].

The taxonomic classification of HMs has not been fully clarified and remains unclear, especially because cultivation is not possible. Initially, HMs were classified within two genera, i.e., *Eperythrozoon* and *Haemobartonella* within the order *Rickettsiales*, family *Anaplasmataceae.* This was followed by a reclassification into the order *Mycoplasmatales*, family *Mycoplasmataceae*, genus *Mycoplasma* due to sequence analyses of the 16S rRNA and RNase P genes [14,15,16,17]. Within the genus *Mycoplasma* with mostly mucosal-associated bacteria, HMs represent a distinct group of erythrocyte-associated organisms. Due to the lack of in vitro cultivation systems, it remains difficult to fully characterize the taxonomic relationship of HMs. Therefore, HMs have been given the status “*incertae sedis*” within the family *Mycoplasmataceae*. In 2018, Gupta and co-workers proposed the reclassification of HMs within a “new” genus *Eperythrozoon*; however, this proposal has not been implemented in practice [18].

Hemotrophic mycoplasma infections in livestock can cause acute and chronic diseases but also asymptomatic infections. Both types of infections are known as a significant cause of economic loss and welfare concern in the agricultural sector worldwide. The economic impact is on the one hand the result of clinical signs caused by the HMs themselves and on the other hand due to the increased susceptibility to other infectious agents caused by the infection-induced immune dysregulation [1,2,19,20].

Transmission of HMs mainly includes procedures exposing or sharing blood such as iatrogenic procedures or ranking fights [1,5]. Also, transplacental transmission has been described [21,22]. Blood-independent infections by excretions play a minor role under field conditions [23,24]. Because HMs reside on or within erythrocytes, there is considerable potential for livestock HMs to be taken up by a variety of hematophagous arthropod vectors such as flies, midges, mosquitoes, ticks, and lice [5]. However, experimental proof of HM transmission by arthropods is rare. This review aims to analyze and summarize published data on the transmission of HMs by arthropod vectors in livestock.

## 2. Clinical Signs and Prevalence of Hemotrophic Mycoplasmas in Livestock

Hemotrophic mycoplasma infections in livestock cause significant economic loss in the agricultural sector worldwide. The economic impact is on the one hand the result of the clinical outcomes and performance failure caused by HM infections, and on the other hand due to the increased susceptibility to other pathogens caused by an HM-induced immune dysregulation [2,4,25]. The immune response often fails to eliminate the pathogens leading to persistence and chronicity. Therefore, prophylaxis and therapy are not easy owing to the common persistence of HM infections and their complex pathogenesis [2,4].

The following chapter provides an overview on published data describing the clinical outcomes and prevalences of the individual HMs in farm animals, specifically the three porcine species *M. suis*, *M. parvum*, and ‘*Ca*. M. haemosuis’, the bovine species *M. wenyonii*, and ‘*Ca.* M. haemobos’, and the small ruminant HMs *M. ovis*, and ‘*Ca*. M. haemovis’ [8,9,10,11,12,13].

### 2.1. Porcine Hemotrophic Mycoplasmas

To date, three HM species are known to infect pigs. *Mycoplasma suis* and *M. parvum* were first described in 1950 by Splitter as *Eperythrozoon suis* and *Eperythrozoon parvum* [8]. Recently, the third porcine HM species ‘*Ca*. M. haemosuis’ was identified in China in 2017 [9].

Most studies deal with *M. suis*, the longest known and most pathogenic representative of the porcine HMs. *Mycoplasma suis* causes infectious anemia of pigs (IAP), which can manifest in various degrees of severity, depending on host susceptibility and virulence [26]. Acute infected pigs show high fever, pallor due to hemolytic anemia, and hypoglycemia. Typical skin alterations can range from petechia or urticaria, acro-cyanosis and necrosis, to generalized skin hemorrhages. Furthermore, acute *M. suis* infections are associated with reproductive disorders in sows including dysgalactia, fertility disorders, and abortion [1,27,28,29,30]. More often, chronic *M. suis* infections with low-grade bacteriemia occur. Clinical signs in chronically infected animals vary from asymptomatic outcomes to a wide range of clinical conditions including anemia, mild icterus, growth retardation in feeder pigs, or poor reproductive performance in sows [26,27,28,31].

Contrary to *M. suis*, there are only a few reports regarding *M. parvum* infections. Although both porcine HM species are genetically remarkably similar and share all coding sequences (CDS) with known functions [32,33], *M. parvum* seems to be low or non-pathogenic and infections are mostly asymptomatic. Even after splenectomy, infected pigs developed either no or only mild clinical signs such as mild anemia and low-grade fever [32]. However, the significance of asymptomatic *M. parvum* infections for pig health has not been fully clarified [34]. Two recently published studies found correlations between *M. parvum* infections and reduced weight gain and slaughter weights in fattening pigs and a reduced number of weaned piglets in clinically healthy sows [35,36].

Recently the third porcine HM species ‘*Ca*. M. haemosuis’ was detected in pigs with and without clinical signs in China, Korea, and Germany [9,37,38]. Sequencing of the 16S rRNA gene demonstrated that ‘*Ca*. M. haemosuis’ is similar to the feline HM species ‘*Ca*. M. turicensis’. Described clinical signs were like those found in *M. suis* infected pigs and include high fever, reduced feed intake, skin alterations, increased mortality, and decreased daily weight gain [38].

Since HMs cannot yet be cultivated and microscopic detection lacks specificity and sensitivity, only molecular studies are considered. Reports on the molecular detection of porcine HMs in blood samples including wild boars are available from several countries all over the world. The found *M. suis* detection rates at single animal level are highly variable, ranging from 10% to 53% in Europe (France, Switzerland, and Germany [25,39,40]), from under 2% in piglets to 76% in sows in South America [41,42,43], and from 0.2% to 86% in Asia (Republic of Korea, China, and Japan; [37,44,45]). At herd levels, the detection rates are still higher. Prevalences ranging from 9% in Japan to 100% in Europe have been reported for pigs [39,44].

Data on the distribution of *M. parvum* infections are rare and mostly on a single animal level. Few PCR-based studies report the occurrence of this species in domestic pigs, among others in South America [35,46], Asia (China, Japan, Republic of Korea, and Thailand [9,44,47,48], and in Europe (Germany) [34]. Additionally, Fernandes et al. detected *M. parvum* in wild boars in Brazil [49]. Prevalences were determined by Ade and colleagues (2022) in different age groups and production types. Interestingly, they found wide variations in the *M. parvum* detection rates with rates of 4% in boars, 25% in sows, and 36% in fattening pigs [34]. Potential explanations for these differences include potentially higher biosecurity levels in boar studs, with individually housed animals, and the absence of group vaccinations in boar studs compared with fattening and piglet-producing farms. In the studies in Japan and South Korea, *M. parvum* determined prevalences ranging from over 2% to 15% in pigs of unspecified age [37,44] were described.

Since ‘*Ca*. M. haemosuis’ is an emerging pathogen, there are currently only few published data on the occurrence of ‘*Ca*. M. haemosuis’ in domestic pigs from Asia (China, Republic of Korea, and Thailand) [9,37,48], as well as from Germany [34], with detection rates of 0% in boars and 36% in sows.

### 2.2. Bovine Hemotrophic Mycoplasmas

At least two distinct HM species were identified in cattle, i.e., *M. wenyonii* and ‘*Ca*. M. haemobos’. *Mycoplasma wenyonii* was first described in a splenectomized calf in 1934 [10] and has since been identified in several countries in cattle, buffalo, and sheep [50,51,52,53,54,55,56]. The second bovine HM species ‘*Ca*. M. haemobos’ was first reported in Japan [57]. The 16S rRNA gene sequencing revealed that the feline HM species *M. haemofelis* is the closest relative to ‘*Ca*. M. haemobos’. The novel bovine HM species was shown to infect cattle, buffalo, and small ruminants [11,52,58,59]. New studies also show a high dynamic of the bovine HM genomes (variable genome sizes, position shifts of genes, the little conserved gene synteny) and strong regional differences, e.g., between Mexican isolates and other American isolates [60].

Various clinical signs have been associated with *M. wenyonii* and ‘*Ca*. M. haemobos’ infections in cattle including hemolytic anemia, pyrexia, reproductive disorders, edema, and decreased production outcomes (e.g., decreased milk production, weight loss) [5,56,59,61]. Occasionally, infected cattle die [59]. Chronic infection with *M. wenyonii* and ‘*Ca*. M. haemobos’ can cause decreased milk production [62]. Anemia, decreased milk production, infertility, and lameness were also described in cattle co-infected with *M. wenyonii* and ‘*Ca*. M. haemobos’ [59,63]. In sheep and goats fever, anorexia, depression, pale oral cavity mucous, eye conjunctiva, hematuria, and lamb mortality were delineated [58]. Although little is known about the pathogenesis of *M. wenyonii* and ‘*Ca*. M. haemobos’, some studies suggest that ‘*Ca*. M. haemobos’ is more pathogenic than *M. wenyonii* [64]. In addition, it was assumed that clinical signs of HM infections in cattle could be more severe if co-infections with both HM species or other pathogens (e.g., *Anaplasma* spp.) occur [63].

Detection and prevalences of bovine HMs have been recently and comprehensively reviewed by De Souza Ferreira and Ruegg in 2024 [54]. Both, *M. wenyonii* and ‘*Ca*. M. haemobos’ have been detected in cattle in several countries all over the world including Europe (Switzerland, Hungary, France, and Germany), Asia (Japan, China), America (USA, Brazil, and Cuba), New Zealand, and Asia (Japan and China). The found prevalences were highly variable on single animal level, ranging from 0% to 95% for *M. wenyonii* and from 2% to 97% for ‘*Ca*. M. haemobos’ [54].

### 2.3. Ovine Hemotrophic Mycoplasmas

Mainly two HM species, i.e., *M. ovis* and ‘*Ca*. M. haemovis’, could be found in small ruminants throughout the world. Mycoplasma ovis can infect sheep, goats, deer, reindeer, and humans. In lambs and young sheep, acute infections can occur with severe hemolytic anemia and decreased exercise tolerance [5,13]. In older animals, *M. ovis* infections are mostly chronic with absence of or mild clinical signs due to the low pathogenic potential [5,65]. Sheep with chronic infections show mild parasitemia and regenerative anemia, ill thrift, and reduced production outcomes (e.g., poor weight gain, decreased wool and milk production) [5,13,66]. In goats, *M. ovis* cause a more severe course of disease [66].

‘*Candidatus* M. haemovis’ was first detected in sheep (mainly yearlings) with fatal hemolytic anemia in Hungary [13,66,67]. In Japan, ‘*Ca*. M. haemovis’ was also detected in yearlings with mild anemia and in older sheep, co-infected with *M. ovis* [66,67]. ‘*Ca*. M. haemovis’ can also affect goats [68].

Ovine HM infections were also detected worldwide. Similar to all other HM species in livestock mentioned before, PCR-based prevalences for the ovine HM representatives also vary considerably [5], mostly on a high level. For example, Urie and co-workers found *M. ovis* prevalences of 69% to 76% in the USA [69]. In Brazil, a much lower *M. ovis* prevalence of 39.3% was detected in goats [70] but a higher prevalence was detected in sheep (79%) [71]. In Europe, Hornok and co-workers determined the rate of HM-positive sheep in Hungary to be 52% [13] and Aktas and Ozubek determined the rate of HM-positive animals in Turkey to be 54% [72]. Considerably lower detection rates were found in Tunisia, where only 6% of the tested sheep and 0% of the goats were *M. ovis* positive [73]. In Asia, HM prevalences (*M. ovis* and ‘*Ca.* M. haemovis’) ranging from 36% to 45% were found in goats and sheep [74,75].

## 3. Vector-Based Transmission of Hemotrophic Mycoplasma in Livestock

The epidemiology of HM infections in livestock is still insufficiently understood, and many questions remain about possible mechanisms of transmission. Figure 1 gives an overview on the transmission pathways described so far.

Since HMs parasitize mainly in the blood of infected animals, it is generally accepted that blood transferring contact between animals during iatrogenic manipulations, or through small wounds due to technopathies or rank fights, represents the main transmission route. However, high prevalences were not only found in conventional low-space stables but also in extensive farming or in grazing animals. Therefore, there are many discussions about a possible transmission of HMs by arthropod vectors. Because of their capability (i) to reside on or within erythrocytes and (ii) to establish chronic and persistent low-grade infections, there is a permanent and substantial opportunity for all HM species to be taken up by a variety of hematophagous arthropods during suckling on a bacteriemic host. Nevertheless, it remains rather unknown whether HMs are only transported by the arthropod from one host to another (mechanical vector) or if HMs could multiply within arthropod vectors and then be transferred to the next host (biological vectors). In addition, most studies discussing the transmission of HMs by arthropods are based on the detection of the pathogens by PCR methods and, therefore, demonstrate vector potential but fail to prove real vector competence, which means that HMs are actually transmitted. Studies demonstrating the real vector competence of arthropods are mostly from the pre-PCR era. The following chapters summarize the current knowledge on the potential transmission of livestock-associated HMs by arthropod vectors such as flies, mosquitoes, and ticks. Older studies performed in the pre-PCR era mainly combined microscopic methods and animal experiments. All studies now use PCR methods to detect HMs in potential vectors. Table 1 summarizes the current knowledge on flies and mosquitoes as vectors, Table 2 gives an overview about the published studies on the transmission of livestock-associated HM species by ticks.

### 3.1. Transmission of Hemotrophic Mycoplasmas by Mosquitoes

There are a few studies investigating whether mosquitoes could transmit HMs from animal to animal. Prullage and co-workers utilized the yellow fever mosquito, *Aedes aegypti*, to determine their capability to transmit *M. suis* between pigs. For this, mosquitoes were fed on *M. suis* infected pigs for 5 to 10 min and then transferred immediately or after 7 days to susceptible splenectomized pigs. A successful transmission was accomplished in all used pigs (*n* = 9) when the transfer of *Aedes aegypti* was carried out immediately. This indicates a high transmission potential and a role of mosquitoes as mechanical vector. However, none of the pigs became infected when there was a delay before transfer to the susceptible pig, indicating that *M. suis* ingested in *A. aegypti* lose their infectivity during the 7-day period of storage in the chamber [76]. Howard (1975) showed that *M. ovis* was transmitted by immediately transferring feeding *Aedes camptorhynchus* from infected to non-infected sheep. A transmission of *M. ovis* was also achieved when the mosquitoes were held for 6 days between feeding on infected sheep and recipient sheep [77]. Another study used *Culex annulirostris* to investigate a transmission of *M. ovis* from sheep to sheep by mosquitoes [78]. Mosquitoes were fed on *M. ovis* infected lambs and then transferred to uninfected animals which then became infected. Additionally, fed mosquitoes were maintained for 6 days and then grounded and extracted. Inoculation of the mosquito extracts led to infections of the lambs, indicating that *M. ovis* remained infective in mosquitoes for at least 6 days. All in all, the studies suggest that many mosquito species possess substantial competence as a mechanical vector for HMs [78].

Other studies used PCR methods to detect HM species in mosquitoes. Song and co-workers utilized a combination of PCR and LAMP methods to determine the occurrence of *M. wenyonii* in mosquitoes collected in a stable with clinically healthy but *M. wenyonii* infected cattle [81]. Overall, 21 out of 26 mosquitoes tested positive for *M. wenyonii*, suggesting that mosquitoes could serve as mechanical vectors for bovine HMs. Another study of Reagan and co-workers also found *M. wenyonii* in wild-caught mosquitoes using PCR [82]. However, molecular detection of HMs in mosquitoes collected in HM-positive herds was not consistently successful. Hornok and colleagues failed to detect HM species in *Aedes* and *Culex* spp. [55] and Thongmeesee and co-workers were not able to detect ‘*Ca*. M. haemobos’ in mosquitoes (mostly *Culex tritaeniorhynchus*) collected by traps and/or sweeping techniques in Thailand on a buffalo farm [83].

### 3.2. Transmission of Hemotrophic Mycoplasmas by Stomoxys Species

Stable flies can regularly be found in livestock stables, where they feed blood from the farm animals one or two times per day, mostly on more than a single animal. Besides direct harmful effects on their hosts (e.g., restlessness, skin lesions, pain, and stress), *Stomoxys* spp. are known to be relevant vectors for the transmission of pathogens [84,93]. Several studies could be found in the database investigating the potential of *Stomoxys* in the transmission of HMs. Prullage and co-workers investigated the ability of *Stomoxys calcitrans* to transmit *M. suis* from pig to pig [76]. Stable flies were allowed to feed on *M. suis* infected pigs and were then transferred immediately to negative recipient pigs. *Mycoplasma suis* was successfully transferred in 3 out of 15 pigs. However, if the transfer of *Stomoxys calcitrans* from infected to uninfected pigs was carried out with a delay of 24 h or more, no transmission could be detected. Similar experiments were performed by Overas in 1969 where *Stomoxys* spp. were utilized to successfully transfer *M. ovis* from infected to uninfected sheep [85]. Both studies [76,85] indicate that the *Stomoxys* spp. could have the competence as mechanical vectors for HMs in pigs and sheep. In addition to these older studies, there are some recent PCR-based studies on the occurrence of HM species in stable flies. Although there was no evidence of experimental HM transmission in these studies, the detection of the pathogens in potential vectors nevertheless provides evidence of a vector potential. One study from Thailand was able to detect HM DNA in 41.41% (53/128) of stable flies (*Stomoxys calcitrans*) at a buffalo farm [83]. Subsequent sequencing of some PCR amplicons revealed *M. wenyonii* as HM species [83]. Hornok and co-workers collected *Stomoxys calcitrans* flies in a stable with *M. wenyonii* and ‘*Ca*. M. haemobos’ positive animals [55]. Both bovine HM species were detected in the flies. *Mycoplasma wenyonii* was found in 6/20 *Stomoxys* samples with a mean copy number of 1.0 × 10^2^. ‘*Candidatus* M. haemobos’ was detected in 1/20 flies with a mean copy number of 1.0. Another PCR-based study investigated the occurrence of porcine HM species in stable flies. Schwarz and colleagues investigated *Stomoxys* spp. from a total of 20 farms by PCR with subsequent DNA analysis and found *M. parvum* and *M. suis* positive flies in 7 farms [84].

### 3.3. Transmission of Hemotrophic Mycoplasmas by Tabanidae Species

Two PCR-based studies investigated the potential of *Tabanidae* spp. as potential vectors for HMs [55,83]. Tabanids are hematophagous ectoparasites, which are seasonally present in many areas. Tabanids are known to be vectors for many animal pathogens [94]. Hornok and colleagues detected M. wenyonii in 2 out of 8 *Tabanus bovinus* and 5 out of 16 T. bromius samples, collected in an extensively kept cattle herd [55]. ‘*Candidatus* M. haemobos’ was only found in *T. bovinus* samples (in two out of eight samples). Mycoplasma wenyonii has also been recorded in *Tabanidae* in Thailand [83]. Due to the lack of animal experiments, transstadial studies and studies with greater amounts of samples, further data need to be obtained to learn more about the potential of *Tabanidae* spp. as vectors for HMs.

### 3.4. Transmission of Hemotrophic Mycoplasmas by Lice

Blood-sucking lice are common ectoparasites of domestic animals and can act as either mechanical or biological vectors of various infectious disease agents [95]. However, there are only a few studies investigating the possible role of hematophagous lice as vectors. Heinritzi performed in vivo transmission experiments in pigs [80]. In this study, it was able to transfer *M. suis* to uninfected splenectomized pigs by means of *Haematopinus suis* which had fed on infected pigs for 40 days. Yet, it remains unclear if an infection would have also been successful using unsplenectomized pigs. Recently, another study from Argentina demonstrated the presence of *M. suis* in lice collected in the field. *Haematopinus suis* samples (15.3%) originating from both domestic and wild boars tested positive for *M. suis*, indicating that *H. suis* could serve at least as a mechanical vector for *M. suis* [79]. Otherwise, there are also studies in which hemotrophic mycoplasmas could not be detected in lice. Hornok and colleagues examined various hematophagous lice from cattle, goats, and pigs including *Haematopinus* spp. and *Linognathus* spp. and were unable to detect HM DNA in any of the ectoparasites examined [96].

### 3.5. Transmission of Hemotrophic Mycoplasmas by Ticks

For some time there has been considerable interest in ticks as potential vectors for hemotrophic mycoplasma species mainly due to the wide availability of molecular detection methods. The literature research has revealed evidence on the occurrence of HMs in ticks in Asia (China and Malaysia), Brazil and Mexico. Most of the published data are based on the PCR detection of HMs in engorged female ticks. However, the significance of PCR detection from engorged female ticks remains difficult to interpret as blood residues are always found in the sucked ticks, which can originate from infected animals.

In central China, where ‘*Ca*. M. haemobos’ epidemics have been confirmed, one PCR study demonstrated that 53% of *Rhipicephalus microplus* ticks were positive for ‘*Ca.* M. haemobos’ [58]. Interestingly, most of the ticks originated from goats and sheep during suckling but ticks have also been collected from the surrounding grassland. From the grassland-derived *B. microplus* ticks, 48% were also positive for ‘*Ca.* M. haemobos’, indicating that these ticks harbor ‘*Ca.* M. haemobos’ before sucking the blood of goats and sheep and that ‘*Ca.* M. haemobos’ might be transmitted via the transstadial route from one tick stage to another [58]. In a second study, ticks (*Haemaphysalis longicornis* and *Rhipicephalus microplus*) were collected from dogs living in backyard farms sharing their living area with ‘*Ca.* M. haemobos’ positive sheep and goats. This study could demonstrate for the first time that dogs could be infected with the bovine HM species ‘*Ca.* M. haemobos’ and that 9% of the *H. longicornis* and 47.3% *R. microplus* harbor ‘*Ca.* M. haemobos’ [89]. *‘Candidatus* M. haemobos*’* positive ticks could be detected from both positive and negative dogs, and the authors concluded that further studies are necessary to evaluate whether ‘*Ca.* M. haemobos’ could be transmitted to alternative hosts by *R. microplus* under natural conditions. In another study, the same research group demonstrated that under experimental conditions, (i) ‘*Ca.* M. haemobos’ is transmitted from adult ticks to their eggs and larval stages, (ii) larval ticks can transmit ‘*Ca*. M. haemobos’ to BALB/c mice during feeding, and that (iii) negative larvae could acquire ‘*Ca*. M. haemobos’ from infected mice [88]. Thus, it could be shown for the first time that *R. microplus* ticks have vector competence and therefore probably play an important role in the epidemiology of HM infections in livestock farming, especially in pasture farming. Although this study produces very interesting results, it remains unclear if infected larval ticks would have also been able to infect cattle. The possible capacity of *Rhipicephalus microplus* as a vector for hemotrophic mycoplasmas was further supported by two other studies from Malaysia and Mexico [90,92]. In Malaysia, 36% of *R. microplus* collected from cattle were *M. wenyonii* positive but ‘*Ca*. M. haemobos’ negative [90]. In Mexico, *M. ovis* was confirmed in *Rhipicephalus microplus* ticks collected from infected sheep during an acute outbreak [92].

In Brazil, one study demonstrated that *Amblyomma sculptum* and *Amblyomma ovale* collected from wild boars also might serve as potential vectors for hemotrophic mycoplasmas [87]. A total of 164 engorged ticks were investigated and 8.69% of these ticks harbored HM, *M. suis*, or *M. parvum* as determined by sequence analyses. Also, *Ixodes ricinus* feeding on a *M. wenyonii* positive cow tested positive in Bosnia and Herzegowina [91,97]. However, there are also some studies which failed to identify any HM species in ticks collected from infected animals or in the surroundings of HM-positive herds. For example, Hornok and colleagues failed to detect HM DNA in various hard tick species (i.e., *Ixodes ricinus*, *Dermacentor reticulatus*, *Haemaphysialis inermis*, and *Dermacentor marginatus*) collected in a cattle herd with a known HM history (from cattle and vegetation) [56]. Similarly, Hofmann-Lehmann and colleagues were unable to detect *M. wenyonii* or ‘*Ca.* M. haemobos’ DNA in *Ixodes ricinus* ticks originating from infected dairy cows [11].

## 4. Conclusions

Knowledge on the transmission of hemotrophic mycoplasmas in livestock is very important, in particular, regarding prophylaxis and control strategies which currently mainly comprise biosecurity measures. This review summarizes the current knowledge on hematophagous arthropods as potential vectors for hemotrophic mycoplasmas in livestock. All suspected arthropod vectors described so far are presented in Figure 2 (cattle), Figure 3 (sheep and goats), and Figure 4 (pigs). 

During recent years, there has been growing interest and evidence for arthropod-based transmission and different arthropods that might serve as potential vectors for HM species.

Molecular surveys of arthropods suggest that various arthropods should be considered as potential vectors. Only one molecular study demonstrated that ticks are able to transfer HMs from animal to animal [88]. The capacity of hematophagous arthropods as mechanical vector is further supported by older studies which have demonstrated that HM infections can be transmitted from infected animals to uninfected recipients. Currently, there is no evidence that hemotrophic mycoplasma species are able to replicate within arthropod vectors and there is only one study evidencing that HMs can be transmitted transstadial in ticks [88]. Proof of vector competency will require more experimental transmission studies with infected arthropods to gain knowledge about arthropod-dependent animal-to-animal transmission by ticks and other arthropod vectors. Seemingly, the vector competence of other arthropods should also be further investigated, for example, using omics methods. Future studies should also include a potential replication of HMs within arthropods and whether HMs are transmitted transovarially in the hematophagous arthropod species. At the moment, there are little data comparing the transmission of HMs by arthropod vectors in different climate regions and livestock holding systems; nevertheless, effective vector control tools should be integrated into current prophylaxis measures, including treatment against ectoparasites and environmental management, to reduce risks of HM infection [97]. Also, arthropod feeding behavior, survival of HMs in different vectors, and thus, the amount of HMs in arthropod vectors need to be considered to obtain more data on the potential of different arthropod vectors to infect animals.

## Figures and Tables

**Figure 1 microorganisms-12-01278-f001:**
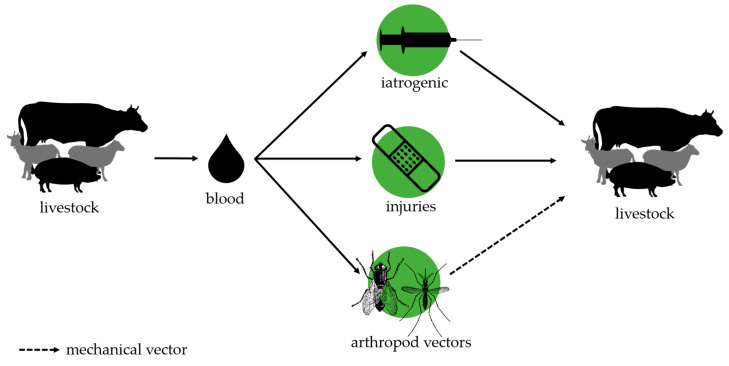
Transmission pathways of hemotrophic mycoplasma in livestock.

**Figure 2 microorganisms-12-01278-f002:**
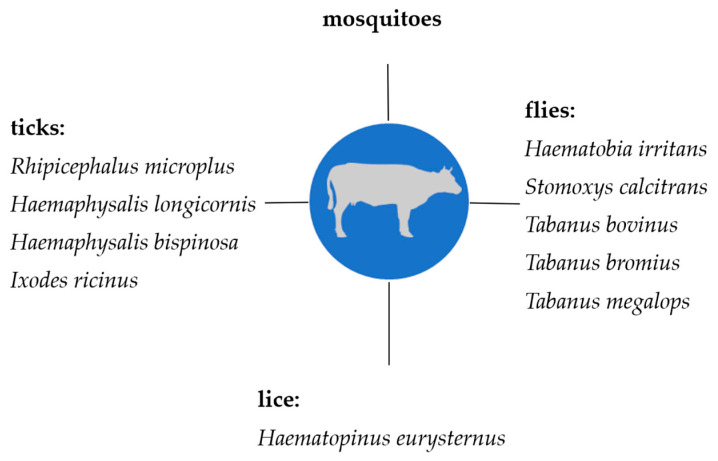
Suspected arthropod vectors for hemotrophic mycoplasmas in cattle described so far.

**Figure 3 microorganisms-12-01278-f003:**
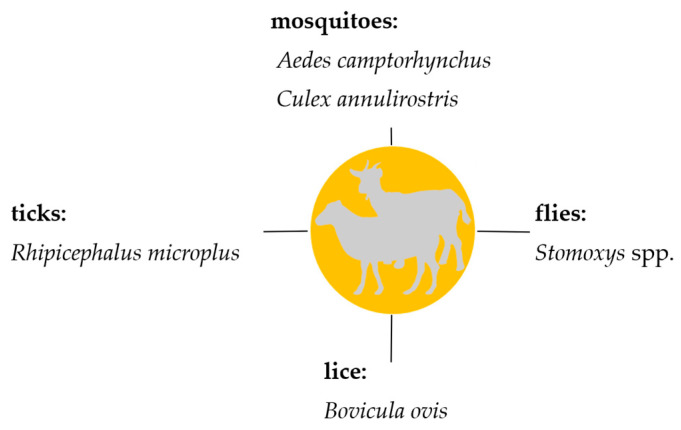
Suspected arthropod vectors for hemotrophic mycoplasmas in goats and sheep described so far.

**Figure 4 microorganisms-12-01278-f004:**
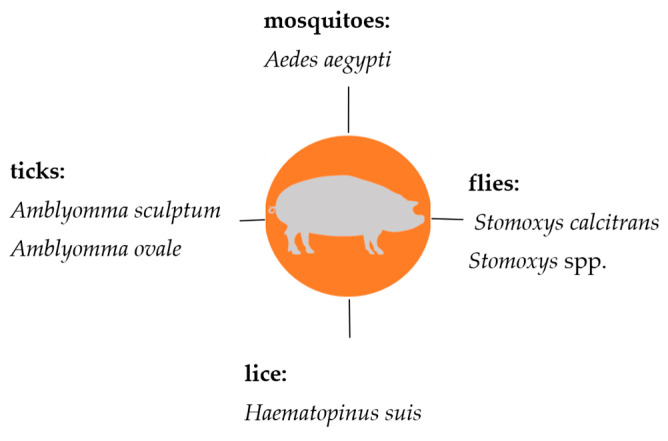
Suspected arthropod vectors for hemotrophic mycoplasmas in pigs described so far.

**Table 1 microorganisms-12-01278-t001:** Known/suspected arthropod vectors for hemotrophic mycoplasma species in livestock (except ticks).

Vector	Methods	HM Species	Results	Ref.
*Aedes aegypti* (mosquito)	Experimental transmission in vivo from infected to uninfected pigs	*M. suis*	Successful transmission in 9 out of 9 pigs	[76]
*Aedes camptorhynchus * (mosquito)	Experimental transmission in vivo from infected to uninfected sheep	*M. ovis*	Successful transmission experiment in sheep	[77]
*Culex annulirostris* (mosquito)	Experimental transmission in vivo from infected to uninfected sheep	*M. ovis*	Successful transmission experiment in sheep	[78]
*Haematobia irritans* (horn fly)	PCR detection	*M. wenyonii* ‘*Ca.* M. haemobos’	Detection of *M. wenyonii* in 1/20 *H. irritans* samples detection of ‘*Ca.* M. haemobos’ in 2/20 *H. irritans* samples	[55]
*Haematopinus suis* (pig louse)	PCR detection and sequence analysis	*M. suis*	Detection of *M. suis* in 15/98 *Haematopinus suis* collected from domestic pigs and wild boars	[79]
*Haematopinus suis* (pig louse)	Experimental transmission in vivo from infected to uninfected pigs	*M. suis*	Successful transmission experiment in sheep	[80]
*Haematopinus eurysternus* (cattle louse)	PCR detection	*M. wenyonii*	Detection of M. wenyonii in 5/5 *H. eurysternus*	[11]
Lice, flies, mosquitoes (without further specification)	LAMP and PCR detection	*M. wenyonii*	Detection of *M. wenyonii* in 18/26 lice, 20/30 flies, and 21/26 mosquitoes	[81]
Mosquitoes	PCR detection	hemotrophic mycoplasma	One pool reacted *M. wenyonii* positive	[82]
*Stomoxys calcitrans* (stable flies)	Experimental transmission in vivo from infected to uninfected pigs	*M. suis*	Successful transmission of *M. suis* to 3/15 pigs	[76]
*Stomoxys calcitrans* (stable flies)	PCR detection	*M. wenyonii* ‘*Ca.* M. haemobos’	Detection of *M. wenyonii* in 7/20 *Stomoxys* samples detection of ‘*Ca.* M. haemobos’ in 1/20 *Stomoxys* samples.	[55]
*Stomoxys calcitrans* (stable flies)	PCR detection	*M. wenyonii*	Detection of *M. wenyonii*	[83]
*Stomoxys* spp. (stable flies)	PCR detection and sequencing	*M. suis* *M. parvum*	*M. suis* and *M. parvum* were detected in *Stomoxys* from 7 out of 20 farms	[84]
*Stomoxys* spp. (stable flies)	Experimental transmission in vivo from infected to uninfected sheep	*M. ovis*	Successful transmission experiment in sheep	[85]
*Tabanus bovinus*, *Tabanus bromius* (flies)	PCR detection	*M. wenyonii* ‘*Ca.* M. haemobos’	Detection of *M. wenyonii* in 2/8 (*T. bovinus*) and 5/16 (*T. bromius*) samples detection of ‘*Ca.* M. haemobos’in 2/8 *T. bovinus* samples.	[55]
*Tabanus megalops* (flies)	PCR detection	*M. wenyonii*	*M. wenyonii* was detected in *T. megalops*	[83]
*Bovicula ovis* *(Mallophaga)*	PCR detection	*M. ovis*	*M. ovis* was detected in one *B. ovis* pool	[86]

**Table 2 microorganisms-12-01278-t002:** Known/suspected tick vectors for hemotrophic mycoplasma species in livestock.

Vector	Methods	HM Species	Results	Ref.
*Amblyomma sculptum* *Amblyomma ovale*	PCR detection	*M. suis* *M. parvum*	8.69% of the ticks collected from wild boars were HM positive	[87]
*Rhipicephalus microplus*	PCR detection and sequence analysis	‘*Ca.* M. haemobos’	70/132 ticks (53%) collected from sheep and goats were positive for ‘*Ca.* M. haemobos’	[58]
*Rhipicephalus microplus*	PCR detection Mice experiments	‘*Ca.* M. haemobos’	Transstadial transmission from positive ticks naturally to egg and larval stage, Transmission from infected larvae to mice Transmission from infected mice to negative larvae	[88]
*Haemaphysalis longicornis* *Rhipicephalus microplus*	PCR detection and sequence analysis	‘*Ca.* M. haemobos’	24/266 *H. longicornis* ticks collected from dogs were ‘*Ca.* M. haemobos’ positive 53/112 *B. microplus* ticks collected from dogs were ‘*Ca.* M. haemobos’ positive	[89]
*Rhipicephalus microplus* *Haemaphysalis bispinosa*	PCR detection	*M. wenyonii*	7/29 *H. bispinosa* ticks were positive 16/44 *B. microplus* ticks were positive	[90]
*Ixodes ricinus*	PCR detection	*M. wenyonii*	8/8 ticks tested positive	[91]
*Rhipicephalus microplus*	PCR detection	*M. ovis*	2 ticks tested positive	[92]

## Data Availability

Not applicable.

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
