# Peer review of "Hemotrophic Mycoplasmas—Vector Transmission in Livestock"

_microorganisms, 2024, doi:10.3390/microorganisms12071278_

Round 1

Reviewer 1 Report

Comments and Suggestions for Authors

Your article comprises a simple enumeration of existing reports on hemoplasms in farm animals. I would have liked to see some sort of analysis of these articles, projections for the future, or something that would move this from a flat review to a review that invites questioning or generating new questions.

Author Response

Please find the reply in the attachment.

Reviewer 2 Report

Comments and Suggestions for Authors

The manuscript, a significant contribution to the field of veterinary research, provides an update on the form of transmission of hemoplasmas of veterinary interest. It offers a comprehensive overview of recent findings from various regions of the world, a topic of great interest to the audience. To further enhance the manuscript, the authors are encouraged to address the following points:

1) It is crucial to ensure the correct usage of italics for the scientific names of the organisms throughout the manuscript. This precision in terminology enhances the manuscript's credibility and aids in the readers' understanding.

2) Consider including the reference for the detection of both hemoplasmas (M. wenyonii and Ca. M. haemobos) in Mexico with the DOI: • 10.26457/recein.v13i52.2183

3) The authors should go deeper into explaining how to know if the strains of hemoplasma identified in Brazil are not the same as those identified in another country in South America. Due to the above, I suggest a phylogeny of hemoplasmas be presented to deepen the discussion of how these pathogens circulate in different geographic regions. See reference DOI: 10.3390/microorganisms10101916

4) In lines 113-114, how do the authors explain the differences in the observed prevalence?

5) The authors should include a section mentioning the omics approaches used to identify hemoplasmas in various vectors. This is, for example, the hemoplasmas that have been identified as part of the microbiome in ticks https://doi.org/10.3389/fcimb.2020.00211

https://doi.org/10.1016/j.crpvbd.2021.100037

https://doi.org/10.1099/mgen.0.000730

6) the authors should review in the following pdf the pathogens of importance for each type of animal to integrate this information into the manuscript

https://rr-africa.woah.org/app/uploads/2023/07/2023-veterinary-mycoplasmas-research-report.pdf

Author Response

(The authors gave the same response as above.)

Reviewer 3 Report

Comments and Suggestions for Authors

The manuscript provides a comprehensive review of the potential transmission routes of hemotrophic mycoplasmas (HMs) in livestock, particularly focusing on the role of arthropod vectors such as mosquitoes, stable flies, tabanids, lice, and ticks. Understanding the transmission pathways of these pathogens is crucial for developing effective control and prevention strategies, which directly impacts livestock health and productivity. However, while the synthesis of existing studies is valuable, the article would benefit from more critical analysis and identification of gaps in current research. With some refinements, the manuscript could be recommended for publication in Microorganisms.

Comments:
I believe this review lacks illustrative material, such as a diagram, that would consolidate the knowledge about the transmission pathways and characteristics of hemotrophic mycoplasmas (HMs) in livestock.

Line 87. "few" should be "a few".

Line 218. "inoculated mosquitoes" or "infected mosquitoes"? Please check.

Lines 227-228. “However, molecular detection of HMs in mosquitoes collected in HM positive herds were not consistently successful [81].” It might be advisable to replace the reference with a more appropriate one or rephrase the sentence because the cited work discusses HM studies in feral cat colonies, not in livestock herds.

Author Response

(The authors gave the same response as above.)

Round 2

Reviewer 1 Report

Comments and Suggestions for Authors

The authors performed the requested corrections.

Reviewer 3 Report

Comments and Suggestions for Authors

I have no comments.